# Metabolome of Exosomes: Focus on Vesicles Released by Cancer Cells and Present in Human Body Fluids

**DOI:** 10.3390/ijms20143461

**Published:** 2019-07-14

**Authors:** Aneta Zebrowska, Agata Skowronek, Anna Wojakowska, Piotr Widlak, Monika Pietrowska

**Affiliations:** 1Maria Sklodowska-Curie Institute—Oncology Center, Gliwice Branch, 44-100 Gliwice, Poland; 2Institute of Bioorganic Chemistry, Polish Academy of Sciences, 44-100 Poznan, Poland

**Keywords:** extracellular vesicles, exosomes, lipids, liquid biopsy, mass spectrometry, metabolomics, metabolites

## Abstract

Exosomes and other classes of extracellular vesicles (EVs) have gained interest due to their role in cell-to-cell communication. Knowledge of the molecular content of EVs may provide important information on features of parental cells and mechanisms of cross-talk between cells. To study functions of EVs it is essential to know their composition, that includes proteins, nucleic acids, and other classes biomolecules. The metabolome, set of molecules the most directly related to the cell phenotype, is the least researched component of EVs. However, the metabolome of EVs circulating in human blood and other bio-fluids is of particular interest because of its potential diagnostic value in cancer and other health conditions. On the other hand, the metabolome of EVs released to culture media in controlled conditions in vitro could shed light on important aspects of communication between cells in model systems. This paper summarizes the most common approaches implemented in EV metabolomics and integrates currently available data on the composition of the metabolome of EVs obtained in different models with particular focus on human body fluids and cancer cells.

## 1. Introduction

Exosomes (EX) are membranous virus-sized (30–150 nm) structures belonging to the group of extracellular vesicles (EVs). This class of vesicles derives exclusively from the inward budding of the endosomal membrane to form the multivesicular body, which fuses with the plasma membrane to release exosomes to the extracellular space [1,2,3]. EVs are a heterogeneous group of vesicles, that in addition to exosomes, include two other major classes: microvesicles (also termed ectosomes) and apoptotic bodies [1,4,5]. Though the classification of EVs based on their biogenesis and cellular origin has been well established in the scientific community, the currently available and commonly used techniques of their isolation do not provide efficient separation of individual classes. Therefore “simplified” nomenclature is accepted nowadays, like small EVs (i.e., <200 nm) and medium/large EVs (>200 nm) [6]. The small EVs class is apparently enriched in exosomes but could also contain a fraction of smaller microvesicles formed by budding of the plasma membrane. In this review term “exosomes” is used afterward for simplicity, yet because of abovementioned limitations of methods used for purification and specification of vesicles that were implemented in quoted papers this should be used read rather as “small EVs”; terms exosomes and (small) EVs are used interchangeably thereafter.

Exosomes are released by many various cell types, including red blood cells, B cells, T cells, mast cells, platelets, endothelial cells, fibroblasts, adipocytes, epithelial cells, muscle, dendritic cells, and tumor cells. Their presence in the extracellular medium (in vitro) and in body fluids (in vivo) is confirmed repeatedly. Exosomes, among others, were found in blood, urine, saliva, breasts milk, ascites effusions, nasal secretions, tears, amniotic, synovial, lymphatic, cerebrospinal, and seminal fluids [7,8,9,10,11,12,13,14,15,16,17,18,19]. Numerous investigations revealed an important role of these vesicles in intercellular communication under both normal and pathological conditions. Exosomes could reach recipient cells in the local environment (paracrine mode) or could be transported to distant tissues via the circulation system (endocrine mode) [8,10,14,20,21,22,23,24]. What is more, recent evidence has shown that in pathological conditions the number of exosomes is significantly increasing comparing to healthy donors. Data showing a high number of these vesicles in bio-fluids of patients with ovarian, prostate, lung, colorectal, and gastric cancers, and acute myeloid leukemia, are available in the literature [25,26,27,28,29,30,31,32]. Moreover, the essential role of exosomes in cancer biology as key mediators of a cross-talk between cancer cells and the immune system cells was revealed, pointing out their crucial involvement in a metastatic cascade [22,23,33,34,35,36,37,38]. Therefore, exosomes are a potential source of tumor biomarkers (e.g., tumor-specific proteins or miRNA) [20,26,27,28,29,30,31,32,39,40,41].

The exosomal cargo consists of selected molecules located inside these vesicles or associated with their membrane [4]. However, the majority of available studies addressed mainly exosomal proteins and RNAs. Metabolites, which are also a part of the exosomal cargo, have been given less attention, so far. Metabolites are defined as (low molecular) end products or intermediates of chemical reactions occurring in the organism. They are varied in terms of a chemical structure and, as a consequence, polarity, lipophilicity, and stability. Classification of metabolites is based on functional groups of molecules. Low-molecular-weight metabolites (LMWMs; size < 900 Da) include alcohols, amides, amino acids, carboxylic acids, and sugars [42]. The second group, often considered as a separate field of analysis, represented by lipids and their derivatives has their own classification, which was comprehensively described by Fahy et al. [43]. As metabolites are representing the intermediate or the end point of any cellular process, they can show phenotype printout of organism state. Therefore, important clinical information on disease stage and response to treatment can be achieved from monitoring metabolic changes in patient’s bio-fluids like blood (whole blood, plasma, or serum), urine, saliva, synovial and cerebrospinal fluid, and semen [44,45,46]. Rising interest in exosome metabolome has been initiated by studies aimed at determination of lipids in membranes of exosomes derived from different cells [7,47,48,49]. Afterward, metabolomics approaches were applied to blood-derived [50,51] and urinary-derived exosomes [52,53], revealing a complex set of molecules. Nevertheless, knowledge of metabolome of exosomes, especially derived from bio-fluids, which have a high clinical value, remains rather limited. Here we aim to summarize the current status of this particular field of metabolomics.

## 2. Methods Used in Studies Oriented on the Metabolome of Exosomes

### 2.1. Exosomes Isolation and Characterization

Both bio-fluids and cell culture medium contain exosomes and could be used for their purification. However, it is important to note that at the same time these specimens contain circulating cells, cell debris, and other classes of extracellular vesicles. Multiple techniques have been used to isolate exosomes from complex mixtures [54,55,56]. Currently, available approaches utilize differences in chemical and physical properties of vesicles in regard to other components of biofluids. Ultracentrifugation (UC), density gradient centrifugation, ultrafiltration (UF), or precipitation are commonly used techniques of exosomes purification that are based on differences in sedimentation and density of different classes of particles [55,57,58]. However, these techniques are recently replaced by approaches based on size exclusion chromatography (SEC) or a combination of SEC with (ultra)centrifugation [54,59,60,61]. Although this method offers advantages for proteomic studies, it has limitations in studies on lipidome because Sepharose CL-4B, the popular SEC matrix (fractionation range of 70–40,000 kDa), may not provide good resolution of (very) small EVs from lipoproteins. Therefore, to enhance the selectivity of exosome capture it is recommended to use immune-affinity techniques (IA), e.g., immunomagnetic beads [62] or nanoplasmon-enhanced scattering (nPES) [63], which are based on immobilized antibodies against specific antigens present in their membranes. For more detailed information on exosomes’ separation techniques authors recommend referring to another review on this issue [64]. The application of abovementioned techniques allows separation of exosomes (small EVs) from “soluble” contaminants and larger vesicles/cell debris. However, the separation of different fractions of exosomes remains a more challenging issue. For example, when tumor-derived exosomes (TEX) circulating in the blood are of particular interest their separation from non-TEX exosomes requires knowledge of specific tumor markers [65].

Taking into account heterogeneity of extracellular vesicles and the necessity of distinguishing them from other objects/vesicles present in body fluids or culture mediums the very important issue is the characterization of isolated vesicles. There are a few commonly used techniques to confirm what type of EVs contain the sample of interest. Biochemical and immune-based methods include Western blotting (WB), flow cytometry (FC), and immunosorbent assays (ISAs), which enables characterization of their molecular content. The size and shape of EVs could be determined by electron microscopy (EM) imaging, atomic force microscopy (AFM), nanoparticle tracking analysis (NTA), photon correlation spectroscopy (PCS), and tunable resistive pulse sensing (tRPS). Limitations and advantages of these methods were summarized by Oeyen et al. [66] and Hartjes et al. [67]. However, it is important to note that none of the abovementioned methods itself is able to cover all needs of vesicle characterization. This is why it is recommended to combine biochemical (molecular) EVs analysis with a high-resolution imaging and other technique revealing the size of analyzed vesicles [6,67].

### 2.2. MS-Based Techniques in Metabolomics Studies

Aforementioned techniques of exosome isolation do not provide complete and absolute removal of “contaminating” components, hence analytical methods enabling discrimination between similar molecules in complex mixtures are required. On the other hand, purification procedures entail an inevitable reduction in exosomes yield and affect the concentration of metabolites, which requires very sensitive methods. The selection of the analytical approach is dictated by the complexity of the analyzed sample and chemical properties of desired compounds. A large diversity of metabolites present in a typical biological sample is the reason why available techniques cannot cover all metabolites in a “single run.” Currently, the most suitable approaches for rapid screening and high-throughput analyses of a broad set of low-concentrated metabolites are tools based on chromatography coupled with mass spectrometry (MS). Therefore, in our paper, we focused on the analysis of metabolic profiles of exosomes using MS technique. This approach offers quantitative identification of hundreds or even thousands of metabolites in the sample. MS-based analyses are widely used for fingerprinting and profiling of metabolites [68,69], but can also be applied to study only a selected class of compounds in the targeted analysis [70]. However, many different analytical techniques based on a combination of chromatography and mass spectrometry exist that could be selected depending on the physical and chemical properties of metabolites of interest. Major types of such techniques are listed in Table 1.

There are two critical elements of any mass spectrometer that determine its applicability for analysis of different compounds: type of ionization and type of mass analyzer. First of all, the capability of identification and quantification of metabolites strongly depends on the ionization process. If labile fragments are present in the structure of an analyte, they can break away from the parental molecule and be ionized as its fragment. The higher fragmentation, the more possible it is to find the unique fragment and identify an unknown compound. This very important criterion is met by electron impact (EI) ionization used in GC-MS analysis of LMWMs and by electrospray ionization (ESI) used in LC-MS/MS for lipids. However, two other ionization types, atmospheric pressure chemical ionization (APCI) and matrix-assisted laser desorption ionization (MALDI), could be also used. APCI is suitable for LMWMs which have a relatively stable structure when exposed to high temperature for vaporization [76], while MALDI for non-volatile, large molecules, for example, lipids [75]. Ionization of the analyte molecules is followed by selection and separation of ions by mass analyzers. There are several types of mass analyzers characterized by different mass accuracy and mass resolution used in metabolomics. In metabolome profiling, as well as in targeted analysis, isomeric and isobaric compounds (molecules having the same *m*/*z*) can occur and are usually difficult to separate on chromatography columns. Analyzers with high mass accuracy and resolution power bring the opportunity to resolve the mixture of such molecules. Isomeric and isobaric LMWM can be well separated on GC-EI-MS with single or triple quadrupole, but in lipidomics, much higher resolution power is required (*m*/*z* accuracy ≤ 5 ppm) [77]. This condition can be fulfilled by using time of flight analyzer (TOF), Fourier-transform ion cyclotron, or ion traps (linear and two-dimension). Ion traps, like Linear Trap Quadrupole (LTQ)-Orbitrap, are also very useful for multiple reaction monitoring (MRM) of metabolite ions chosen in targeted analyses [78]. However, despite all these improvements, the information about lipid subclass is frequently missed. This issue can be addressed using ion mobility spectrometry (IMS) which cope with stereomeric diversity of metabolites [79].

GC-MS is frequently considered a ‘gold standard’ in quantitative metabolomics due to its high sensitivity [68]. However, the application of GC-MS is limited to volatile and thermally stable molecules. To increase the number of analytes that could be targeted by GC-MS, chemical derivatization is required. In this strategy, silylation of hydroxyl and primary amine groups along with oximation of carbonyl groups results in reduced polarity and increased volatility of analytes [74,80]. GC-MS is suitable for analysis of LMWMs from different classes, including sugars, carboxylic acids, amino acids, alcohols, and amines. Lipids cannot be analyzed comprehensively with this technique as they have different chemical features (higher molecular weight, polar, and non-volatile). The information on lipids in a wide range of masses can be achieved by LC-MS [81,82]. The advantage of LC-MS is that it can also be preceded by derivatization to increase signals of low-abundant and poorly ionizing metabolites, such as thiol compounds [83]. Recently, thin layer chromatography coupled with MS has been proposed as an alternative tool in lipidomics but only for separation and detection of lipids with *m*/*z* > 500 [75]. However, this technique provides data for lipid classes identification without precise information on subclasses, which is the bottleneck in lipidomics. Each MS-based technique has its advantages but also limitations. MS-coupled systems can be modified to overcome some technical obstacles. Therefore, different analytical approaches based on MS should be taken into consideration in studies that address the metabolome of exosomes; these approaches are schematically illustrated in Figure 1.

## 3. Metabolites Present in Exosomes—Results of Metabolomics Studies

Although the general profile of blood (serum and plasma) or urine metabolites has been investigated with promising results in many studies [84,85,86,87], metabolomics of exosomes is a new approach and knowledge of metabolites present in extracellular vesicles has started to accumulate in a few recent years. A large part of relevant studies took under investigation exosomes (small EVs) derived from the cell culture medium, while the ones addressing vesicles derived from body fluids are less represented. Based on these studies it has been established that exosomes contain different classes of both low-molecular-weight compounds (small molecules) and lipids. Most of the analyses are focused on EVs lipidome and report detection and quantification of different classes of lipids, including glycerolipids (GL), glycerophospholipids (GP), sphingolipids (SP), sterol lipids (ST), prenol lipids (PR), and fatty acids (FA) [7,11,47,52,88,89,90,91,92,93,94,95,96,97]. Part of the studies showed a similarity between the lipid content of EVs and the composition of their parental cells’ membranes, while others reveal disparities in proportions of specific membrane components. Moreover, the available data indicated marked differences in lipid composition of EVs derived from different cells (e.g., ceramide enrichment in EVs produced by tumor cells, but its absence in EVs from RBCs) [7,47,88,95]. Much fewer papers addressed a complete set of metabolites present in EVs and showed that their metabolome contains not only lipids, but also organic acids, amino acids, sugars and their conjugates, nucleotides and nucleosides, cyclic alcohols, carnitines, aromatic compounds, and vitamins [50,51,98,99,100]. The selection of available literature in the area of EV metabolome is listed below (Table 2).

### 3.1. Analysis of Metabolome in Small EVs Derived from Cell Culture Medium in Vitro

Exosomes play an essential role in communication between cancer cells and their microenvironment, which is visible also at the level of their metabolome. Zhao et al. [100] showed that exosomes released by cancer cells and cancer-associated fibroblasts (CAFs) participate in the regulation of cancer cells metabolism. Downregulation of mitochondrial activity and increased glucose uptake and glycolysis was observed in PC3 cancer cells cultured in the presence of EVs derived from CAFs from prostate cancer (PCa) patients. Increased level of pyruvate and lactate was correlated with a reduced level of α-ketoglutarate, fumarate, malate, and glutamate, which confirmed that EVs derived from CAFs stimulated the Warburg effect in recipient cancer cells. Moreover, these authors revealed high levels of different amino acids, carboxylic acids, and fatty acids in EVs released by different types of CAFs, and postulated that EVs act as carriers of metabolites enabling intensification of cancer cell metabolism [100]. Similarly, the presence of lactic and glutamic acids in EVs released by serum deprived mesenchymal stromal cells supported a potential role of EVs in the regulation of glucose metabolism [96]. Therefore, EVs released to culture media by specific cell types in vitro are an interesting model to study mechanisms related to the regulation of cancer cell metabolism.

The complete composition of different classes of metabolites in EVs purified from cell culture media was performed for PANC1 cells derived from pancreatic cancer. This analysis showed that major components of EVs metabolome are main components of membranes: glycerophospholipids and sphingolipids (they comprised 56% of detected compounds), fatty acids esters, amides and alcohols (14%), nucleotides and derivatives (7%), amino acids (6%), and eicosanoids, steroids and prenols (5%). Sugars, cyclic alcohols, aromatic compounds, and organic acids were also found, but less represented [50]. Other studies on metabolites detected in EVs purified from culture media focused on their lipid profile. Llorente et al. [91] compared lipidome of EVs and parental PC3 cells; there were 217 and 250 lipids detected, respectively, with 190 species common for both types of samples. The study showed enrichment of EVs in cholesterol, sphingomyelins, glycosphingolipids, and phosphatidylserines [91]. Hosseini-Beheshti et al. [92] performed an analysis of lipid content of EVs and parental cells for six different prostate cell lines. In general, they observed differences between cells and their EVs in the relative abundance of glycerophospholipids (average content 86.3% and 65.1%, respectively) and sphingolipids (9.6% and 30.2%, respectively). Moreover, cells derived from prostate cancer contain significantly less cholesterol than cells from benign prostate (RWPE-1), while the average cholesterol content of EVs derived from PCa cells was three times higher than EVs derived from RWPE-1 cells [92]. Increased level of sphingolipids and cholesterol in cancer-derived EVs was confirmed in other models. Lydic et al. [93] revealed a higher concentration of sphingomyelins, phosphatidylserines, and cholesterol in EVs derived from colorectal cancer cell line LIM1215 while compared to parental cells. There were different proportions of glycerophospholipids (91.5% vs. 68.3%), sphingolipids (5.3% vs. 22.7%), sterol lipids (1.9% vs. 4.3%), and glycerolipids (1.4% vs. 4.8%) in parental cells and their EVs, respectively [93]. Elevated levels of sphingolipids, phosphatidylserines, phosphatidylethanolamines, and phosphatidylglycerols, while decreased levels of phosphatidylcholines were observed in EVs derived from melanoma FEMX-I cells [94]. Sphingomyelin, ceramides, glycolipids, free fatty acids, phosphatidylserines, and cholesterol were enriched in EVs releases by glioblastoma cells (U87), hepatocellular carcinoma cells (Huh7) and bone marrow-derived MSCs [95]. The high abundance of sphingomyelins and ceramides was observed also in EVs released by serum-deprived MSCs [96], while sphingomyelin and cholesterol enrichment was found in EVs derived from rat mast cells (RBL-2H3 cells) [7], and human B-cells (RN HLA-DR15^+^ cells) [48]. Therefore, the increased level of sphingomyelins and cholesterol, and the decreased level of glycerophospholipids (especially phosphatidylcholines) appeared a general feature of EVs when compared to the lipid composition of their parental cells.

### 3.2. Composition of The Metabolome in Small EVs Derived from Body Fluids In Vivo

The metabolome and lipidome of EVs derived from blood and urine are the most often investigated in this field of research. Studies based on different LC-MS approaches revealed that the major class of lipids detected in plasma-derived EVs (pEVs) and urine-derived EVs (uEVs) comprised of glycerophospholipids and sphingolipids, i.e., major components of membranes. Moreover, the metabolome contained fatty acids and amino acids, steroids, prenols and eicosanoids, peptides and peptide conjugates, nucleotides, nucleosides and their derivatives, as well as less abundant sugars, alcohols, amino acids, and carboxylic acids [50,51,98,99]. Nevertheless, all these classes of metabolites present in EVs derived from human bio-fluids have high importance as a potential source of disease biomarkers.

Several metabolome-oriented studies addressed the composition of EVs derived from the urine of patients with genitourinary malignancies. Puhka et al. [98] studied metabolome of EVs derived from urine and plasma of prostate cancer (PCa) patients and detected 102 metabolites in uEV and 111 metabolites in pEV samples. There were 11 metabolites specific for uEV (creatinine, l-cystathionine, gamma-glutamylcysteine, guanidynoacetic acid, 4-hydroxyproline, kynurenic acid, glucuronate, pantothenic acid, 4-pyridoxic acid, 1-methylhistamine, trimethylamine N-oxide), and five metabolites specific for pEV (kynurenine, lysine, threonine, tryptophan, cytidine). Moreover, several metabolites showed markedly different abundances between uEV and pEV. This is noteworthy that this work revealed differences in uEV metabolome content between cancer patients before prostatectomy and after prostatectomy, as well as between untreated cancer patients and control group. There were four metabolites with a lower level in the pre-prostatectomy group: adenosine, glucuronate, isobutyryl-L-carnitine, and D-ribose 5-phosphate. The largest difference was noted for glucuronate (20-fold difference) between untreated cancer patients and treated patients combined with a control group [98]. Another study of a complete metabolome profile of EVs derived from the urine of prostate cancer patients and patients with benign prostate hyperplasia (BPH) was reported by Clos-Garcia et al. [99]. There were 248 metabolites detected out of which 76 showed significant differences between BPH and PCa patients. Moreover, there were five molecules, namely ceramides Cer (d18:1/16:0), Cer (d18:1/20:0), Cer (d18:1/22:0), PC (30:0), and acylcarnitine AC (18:0) expressed differentially between two subgroups of PCa patients (stages 2 and 3) [99]. Another two studies addressed the lipid content of EVs derived from the urine of PCa patients. One of them identified 36 lipid species and revealed that few of them, including lactosylceramide LacCer (d18:1/16:0), phosphatidylserine PS (18:1/18:1), and PS (16:0–18:1), showed markedly different levels between uEV from cancer patients and healthy controls [89]. Yet another study identified 286 lipids in uEV from PCa patients and healthy controls and showed that several classes of lipids (except for diacylglycerol, triacylglycerol, and cholesterol esters) were more abundant in uEV of cancer patients [52]. Lipid profile of uEV was analyzed also in a group of patients with renal carcinoma, which showed several differences between cancer patients and healthy controls [90]. A few studies addressed the metabolome composition of EVs derived from the blood of patients with other malignancies. There were about 1950 metabolites detected in EVs derived from serum of patients with pancreatic cancer (PANC). The analysis revealed several metabolites, including alanyl-histidine, 6-dimethyl-aminopurine, leucylproline, and methionine sulfoxide, whose abundances differentiated samples collected before and after chemotherapy [51]. The comparison of a metabolic profile of EVs derived from plasma of patients with endometrioid adenocarcinoma and healthy controls also revealed several discriminatory compounds [50]. Very few metabolome-oriented studies addressed EVs present in other human bio-specimens and related to not cancer health conditions. Brouwers et al. investigated a lipid profile of EVs derived from the seminal fluid of vasectomized men [11]. A lipid profile of EVs derived from placental syncytiotrophoblast of pregnant women was analyzed by Baig et al. [97]. The study revealed several EV lipids whose levels differentiated women with preeclampsia or history of recurrent miscarriages from women with a healthy pregnancy.

## 4. Conclusions

Extracellular vesicles—their composition, biology, and role in the pathophysiological processes—are extensively studied. However, the variability of data available in the literature is mostly concentrated on their proteome and transcriptome (miRNome in particular). Studies on the metabolic profile of exosomes are the youngest part of this area. Nevertheless, from the practical and clinical point of view, metabolomics of exosomes derived from human bio-fluids is the most appropriable because the metabolome of these vesicles is a potential goldmine of disease biomarkers. Unfortunately, despite a promising start of metabolomics of exosomes derived from in vitro cell models, much less information is available concerning the metabolome of body fluids-derived vesicles. Results available in literature until now give the first input into this knowledge, but they still need to be extended and validated.

It is also important to understand the future challenges of exosome metabolome studies. The relevant analytical pipeline should consist of separation techniques for efficient and specific exosomes purification, instrumental analysis for a sensitive and specific measurement of metabolites, and adequate data processing. The concerns of all studies with EVs are standardization and improvement of methods of isolation and distinguishing of EV subpopulations. Additionally, one of the concerns of exosomes derived from body fluids is the reduction of contamination from lipoproteins and “soluble” compounds. Another important issue addressed to metabolomics of exosomes is the detection of low abundant metabolites that could be solved with targeted MS analyses if relevant compounds were identified in the source material and their transitions (ions specific for the precursor and the product of each metabolite) are known. For identification of metabolites present in bio-fluids, Human Metabolome Database (HMDB) [101] is a very useful tool. Unfortunately, identification of exosomal metabolites based on currently available databases dedicated for extracellular vesicles is less feasible. Open access databases for exosome small metabolites (EVpedia) and lipids (EVpedia, ExoCarta) do not offer comprehensive information that could be compared to the knowledge of exosome proteins and RNAs.

One should admit the lack of widely accepted gold standard of exosome metabolome analysis at present. Diversity of exosome sources, isolation methods, and analytical techniques together with a limited amount of research performed, foreclose the possibility of rational and well-balanced comparison of available metabolomics approaches. Although, based on recently published data and own research we could recommend two strategies. A combination of SEC with UC or UF might be particularly suitable for in vitro studies, where an amount of material for EV isolation is not a limiting factor; a specific mass spectrometry approach (e.g., type of spectrometer) is not a critical factor. Another situation concerns work with actual clinical samples, where available amount of material for exosome isolation might be a limiting factor. In that case we recommend implementation of a highly efficient one-step method for exosome isolation (e.g., based on SEC) and targeted MS approach for analysis of pre-selected metabolites. Nevertheless, implemented methodology has to be optimized and tailored to the needs of specific research models.

Nonetheless, the knowledge on metabolites carried by exosomes, especially those produced by cancer cells, accumulates constantly. Therefore, it may soon provide valuable information on the phenotype of cancer cells and provide new biomarkers for disease detection, monitoring, and prognosis.

## Figures and Tables

**Figure 1 ijms-20-03461-f001:**
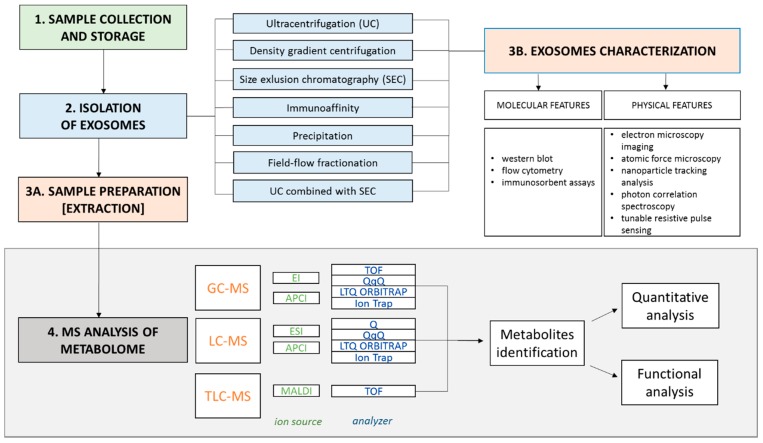
Different analytical approaches combining chromatography and mass spectrometry that could be implemented in studies on exosomes’ (small extracellular vesicles (EVs)) metabolome.

**Table 1 ijms-20-03461-t001:** Selected mass spectrometry (MS)-based techniques applied for metabolomics.

Technique	Abbreviation	Application	References
Gas chromatography-mass spectrometry	GC-MS	Selective method for measuring volatile, thermally stable LMWMs. Requires derivatization of non-volatile LMWMs. Allows the highest sensitivity (in range of pmol–nmol).	[71,72]
Liquid chromatography-mass spectrometry	LC-MS	Measurement of both lipophilic and amphiphilic metabolites with use of different columns on the same device. Covers broad set of metabolites by measuring in negative and positive ion modes.	[73,74]
Thin layer chromatography-mass spectrometry	TLC-MS	An alternative method to direct matrix-assisted laser desorption ionization (MALDI) desorption of sample in lipidomics. Provides glycosphingolipids (GSL) and phospholipids (PL) separation and allows to categorize them into classes.	[75]

**Table 2 ijms-20-03461-t002:** Review on metabolomics studies oriented on the composition of small EVs from different sources.

Cell Type/Body Fluid	Method of EVs Purification and Characterization	MS Approach	Detected Groups of Metabolites	Ref.
Human dendritic cells and RBL-2H3 mast cells	UC/EM	HPLC-MS	lipids (FA, GL, GP, SP, ST)	[7]
Seminal fluid: vasectomized men	UC + SEC/NTA, Cryo-EM, WB	APCI-HPLC-MS	lipids (GP, SP, ST)	[11]
Human B cells (RN HLA-DR15^+^)	UC + SEC/EM, WB	Q-TOF TLC-MS; MALDI-TOF-TLC-MS	lipids (FA, GL, GP, SP, ST)	[48]
PANC1 cancer cell line; human plasma: endometrioid adenocarcinoma patients and healthy controls	UC/NTA, WB	UPLC-ESI-Q-TOF-MS	lipids (FA, GP, SP, ST, PR), carboxylic acids, amino acids, peptides, sugars, cyclic alcohols, nucleotides, and nucleosides	[50]
Human urine: prostate cancer patients and healthy controls	UC/NTA, EM, WB	QqQ ESI LC-MS	lipids (GL, GP, SP, ST)	[89]
Human urine: prostate cancer patients and healthy controls	FFF/TEM, WB	nUPLC-ESI-MS	lipids (GL, GP, SP, ST)	[52]
Human urine: renal carcinoma patients and healthy controls	UC/WB	microLC Q-TOF MS	lipids (GP, SP)	[90]
PC-3 cell line (prostate cancer)	UC/EM, WB	QqQ ESI LC-MS; UHPLC-MS	lipids (FA, GL, GP, SP, ST)	[91]
PC3, DU145, VCaP, LNCaP, C4–2, and RWPE-1 cell lines (prostate cancer)	UC/TEM, WB	UPLC-MS	lipids (GL, GP, SP, ST)	[92]
LIM1215 cell line (colorectal cancer)	UC/TEM, WB	nESI-LC-MS; HCD-MS/MS	lipids (FA, GL, GP, SP, ST)	[93]
FEMX-I cell line (melanoma)	DC + IA/NTA, WB, FM	ESI-LC-MS/MS	lipids (GL, GP, SP)	[94]
U87, Huh7, and hMSCs cell lines (glioblastoma and hepatocellular carcinoma)	UC/NTA, EM, WB	TripleTOF LC-MS/MS	lipids (FA, GL, GP, SP, ST, PR)	[95]
Normal mesenchymal stromal cells (SD-hMSC)	UC/NTA, TEM, IEM, WB	HPLC-MS/MS; LC-MS; SFC-MS	lipids (GL, GP, SP), carboxylic acids	[96]
Syncytiotrophoblast cells (preeclampsia or history of recurrent miscarriage and healthy controls)	UC/NTA, EM	APCI-HPLC-MS/MS; ESI-LC-MS MRM	lipids (GP, SP, ST)	[97]
Human serum: pancreatic cancer patients (before and after CT)	UC/TEM, WB	CIL nLC-MS	lipids (FA, GL, GP, ST), carboxylic acids, amino acids, peptides, biogenic amines, nucleotides, and nucleosides	[51]
Human plasma and urine: prostate cancer patients (before and after prostatectomy) and healthy controls	UC/NTA, EM, WB	TQ-S-UPLC-MS	carboxylic acids, amino acids, sugars, carnitines, biogenic amines, vitamins, nucleotides, and nucleosides	[98]
Human urine: prostate cancer patients and benign prostate hyperplasia patients	UC/NTA, cryo-EM, WB	UHPLC-MS	lipids (FA, GL, GP, SP, ST), amino acids, carnitines, vitamins, nucleotides, and nucleosides	[99]
PCa hCAFs, CAF-35, CAF-19, BxPC3, and MiaPaCa-2 cell lines (prostate cancer and pancreatic cancer)	UC/NTA, FM, WB	GC-MS; UPLC-MS	FA, carboxylic acids, amino acids	[100]

Abbreviations: methods of EVs purification: UC—ultracentrifugation, SEC—size-exclusion chromatography, DC—differential centrifugation, IA—immune-affinity techniques, FFF—field-flow fractionation; methods of EVs characterization: NTA—nanoparticle tracking analysis, EM—electron microscopy, TEM—transmission electron microscopy, IEM—immuno-electron microscopy, FM—fluorescence inverted microscope, WB—Western blotting; lipid classes: FA—fatty acids, GL—glycerolipids, GP—glycerophospholipids, SP—sphingophospholipids, ST—sterol lipids, PR—prenol lipids.

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
