# Peer review of "Metabolome of Exosomes: Focus on Vesicles Released by Cancer Cells and Present in Human Body Fluids"

_ijms, 2019, doi:10.3390/ijms20143461_

Round 1
Reviewer 1 Report
This review is well written and summarize well the current knowledge in metabolomic studies of EVs.
Table 2 is extremely useful. It could be improved by adding one more column, specifying if the study was performed on healthy cells or specifying to which the disease it relates to.
Though the review is manly centred on cancer conditions only – this should be reflected in the title.
Author Response
Thank you for your revision of our manuscript IJMS-540. Enclosed is a revised manuscript, corrected according to your comments. A point-by-point response to specific questions is given below.
Q1. Reviewer #1: Table 2 is extremely useful. It could be improved by adding one more column, specifying if the study was performed on healthy cells or specifying to which the disease it relates to.
A1. In the revised manuscript Table 2 was corrected accordingly. More detailed information on experimental model was added to clarify type of material used in a particular study (type of cells and disease).
Q2. Though the review is manly centred on cancer conditions only – this should be reflected in the title.
A2. The title was changed accordingly to reflect the major focus of the review: “Metabolome of exosomes: focus on vesicles released by cancer cells and present in human body fluids”.
Reviewer 2 Report
Submitted review represents interesting contribution and summary focused on molecular contents of exosomes, mainly in relation to metabolome. I have only several minor concerns:
The title should be more precise, since authors present only results humans and laboratory animal models, mostly associated with cancer or immune cells. There are a number of papers presenting results on exosomes isolated form medically important parasites. Therefore the title and abstract should reflect the content of the article or authors should include more data in their submission.
What concerns the methods used in studies on the metabolome of exosomes. Can you conclude which method or combination of methods seems currently to be the best solution? What about ultrafiltration and nPES, do these techniques have any potential in metabolome research? Similarly in MS identification what about SALD-IMS, FTICR?
This would be beneficial if you could conclude any section and point the paths necessary for solving existing problems.
Author Response
Thank you for your revision of our manuscript IJMS-540. Enclosed is a revised manuscript, corrected according to your comments. A point-by-point response to specific questions is given below.
Q1. The title should be more precise, since authors present only results humans and laboratory animal models, mostly associated with cancer or immune cells. There are a number of papers presenting results on exosomes isolated form medically important parasites. Therefore the title and abstract should reflect the content of the article or authors should include more data in their submission.
A1. The revised title was changed accordingly: “Metabolome of exosomes: focus on vesicles released by cancer cells and present in human body fluids”. Moreover, the focus of the review was underlined in the revised abstract.
Q2. What concerns the methods used in studies on the metabolome of exosomes. Can you conclude which method or combination of methods seems currently to be the best solution? What about ultrafiltration and nPES, do these techniques have any potential in metabolome research? Similarly in MS identification what about SALD-IMS, FTICR?
A2. According to the reviewer suggestions information on nPES and ultrafiltration methods was added to the revised manuscript [section 2.1] together with relevant references. In our best knowledge SALDI-IMS and FTICR mass spectrometry were not used in exosomes metabolomics studies focused on cancer and other human diseases so far.
Q3. This would be beneficial if you could conclude any section and point the paths necessary for solving existing problems.
A3. Additional paragraph was placed in the revised “Conclusions” to address this point.